# Steps Towards Alcohol Misuse Prevention Programme (STAMPP): a school-based and community-based cluster randomised controlled trial

Michael McKay,[1,2] Ashley Agus,[3] Jonathan Cole,[2] Paul Doherty,[3] David Foxcroft,[4] Séamus Harvey,[1,5] Lynn Murphy,[3] Andrew Percy,[6] Harry Sumnall[1]

[1]Public Health Institute, Liverpool John Moores University, Liverpool, UK
[2]Department of Psychological Sciences, University of Liverpool, Liverpool, UK
[3]Northern Ireland Clinical Trials Unit, The Royal Hospitals, Belfast, UK
[4]Psychology and Public Health, Oxford Brookes University, Oxford, UK
[5]School of Sport, Health and Exercise Sciences, University of Bangor, Bangor, UK
[6]School of Social Sciences, Education, and Social Work, Queen's University Belfast, Belfast, UK

**Correspondence to**
Dr Michael McKay;
Michael.McKay@liverpool.ac.uk

## ABSTRACT

**Objectives** To assess the effectiveness of a combined classroom curriculum and parental intervention (the Steps Towards Alcohol Misuse Prevention Programme (STAMPP)), compared with alcohol education as normal (EAN), in reducing self-reported heavy episodic drinking (HED) and alcohol-related harms (ARHs) in adolescents.

**Setting** 105 high schools in Northern Ireland (NI) and in Scotland.

**Participants** Schools were stratified by free school meal provision. Schools in NI were also stratified by school type (male/female/coeducational). Eligible students were in school year 8/S1 (aged 11–12 years) at baseline (June 2012).

**Intervention** A classroom-based alcohol education intervention, coupled with a brief alcohol intervention for parents/carers.

**Primary outcomes** (1) The prevalence of self-reported HED in the previous 30 days and (2) the number of self-reported ARHs in the previous 6 months. Outcomes were assessed using two-level random intercepts models (logistic regression for HED and negative binomial for number of ARHs).

**Results** At 33 months, data were available for 5160 intervention and 5073 control students (HED outcome), and 5234 and 5146 students (ARH outcome), respectively. Of those who completed a questionnaire at either baseline or 12 months (n=12 738), 10 405 also completed the questionnaire at 33 months (81.7%). Fewer students in the intervention group reported HED compared with EAN (17%vs26%; OR=0.60, 95% CI 0.49 to 0.73), with no significant difference in the number of self-reported ARHs (incident rate ratio=0.92, 95% CI 0.78 to 1.05). Although the classroom component was largely delivered as intended, there was low uptake of the parental component. There were no reported adverse effects.

**Conclusions** Results suggest that STAMPP could be an effective programme to reduce HED prevalence. While there was no significant reduction in ARH, it is plausible that effects on harms would manifest later.

**Trial registration number** ISRCTN47028486; Post-results.

### Strengths and limitations of this study

► All data are longitudinal.
► The sample size was very large and attrition relatively low.
► Schools were independently randomised.
► Some of those involved in fieldwork were not blind to participant condition.
► Overall levels of alcohol-related harm were low.

the amount and frequency of consumption increases.[1] Research has shown that family socialisation factors such as approval of adolescent drinking and the provision of alcohol in the home predict drinking among adolescents and young adults.[2–4] An earlier onset of self-reported drunkenness and the establishment of regular alcohol drinking is associated with a greater risk of alcohol-related problems in adulthood.[5] There are also clear geographic and socioeconomic differences in the burden alcohol places on the population, and these are closely associated with other major indicators of ill health and health inequalities.[6–8]

Previous literature reviews have highlighted a lack of high-quality trials of universal school-based alcohol prevention programmes, and few approaches studied have shown positive intervention effects.[9–15] However, while reviews have been unable to recommend any single prevention initiative, many have concluded that interventions that develop social skills appear to be superior to those that seek to enhance only knowledge.[10–13] Guidance issued by the National Institute for Health and Care Excellence in the UK in 2007 called for partnerships between schools and other stakeholders in efforts to prevent misuse.[16] Reviews of universal alcohol prevention in family settings suggest that activities supporting parenting skills, including

## INTRODUCTION

Adolescence is a period when young people experiment with alcohol, and as they age,

establishing clear boundaries or rules and parental monitoring, may be effective.[9 17–19] Primary studies also suggest that when combined with a school-based alcohol curriculum, provision of advice to parents about setting strict rules around alcohol consumption reduces adolescent drinking.[20 21] Indeed, a recently published systematic review reported that of 10 identified combined child-based and parent-based interventions, 9 had reported significant and lasting positive effects on adolescent substance use.[22]

The Steps Towards Alcohol Misuse Prevention Programme (STAMPP) intervention combined a culturally adapted intervention based on the School Health and Alcohol Harm Reduction Project (SHAHRP)[23] curriculum with a researcher-developed brief parental intervention based on the Swedish Örebro Prevention Program.[24] SHAHRP is an example of a resistance skills training programme and includes elements of alcohol-specific personal and social skills training.[25–28] In accordance with the theoretical assumptions underlying such programmes, it includes three main strategies: (1) teaching students to recognise high-risk situations, (2) increasing the awareness of external influences on behaviour and (3) combining self-control (ie, the ability to control responses, to interrupt undesired behavioural tendencies and refrain from acting on them) with refusal skills training (ie, in order to improve self-efficacy in avoiding unhealthy behaviours, but not with the consequence of social disadvantage for the young person with their peers). The knowledge delivered through SHAHRP (eg, lessons on effects of alcohol and description of alcohol units) was not assumed to have direct preventative effects but instead hypothesised to shape alcohol attitudes and support situation-specific decision making. The parental component was based on research indicating that restrictive parenting practices (eg, monitoring of children's alcohol use, healthy attitudes towards alcohol and alcohol rule-setting) was associated with reduced prevalence of children's alcohol use.[21] When this approach was delivered alongside a classroom intervention in the Dutch Prevention of Alcohol Use in Students (PAS), programme effect was mediated through children's perceptions of parental rules, child self-efficacy and child self-control.[29]

It was hypothesised that fewer students in schools delivering STAMPP would self-report: (1) past 30-day heavy episodic drinking (HED) at final follow-up (33 months from baseline) and (2) fewer self-reported alcohol-related harms (ARHs) at final follow-up than those in schools delivering alcohol education as normal (EAN). These primary aim of the research trial were to assess whether STAMPP was effective in reducing self-reporting of these two indicators of alcohol misuse.

## MATERIALS AND METHODS
### Study design
This was a cluster randomised controlled trial (cRCT) of school children in Northern Ireland (NI) and Glasgow/

Inverclyde Education Authority (Scotland) areas in the UK with schools as the unit of randomisation. The trial protocol is available from http://www.nets.nihr.ac.uk/projects/phr/10300209.

### Participants
The sampling frame comprised all mainstream postprimary schools in NI (excluding those within the Eastern Health Board due to existing delivery of SHAHRP in that area) and in Glasgow/Inverclyde Local Authorities. All schools in the sampling frame were assessed for satisfaction of the inclusion criteria and willingness to participate in the trial.

A total of 105 schools were invited to participate in the trial, and all accepted: 70 in NI, 30 in Glasgow Local Authority and 5 in Inverclyde Local Authority. Inclusion criteria were schools in NI and Scotland that taught students in school year 8/S1 in the academic year 2011/2012 (aged 11/12 years at randomisation). Exclusion criteria were schools that did not include students in the specified school year, or only provided non-mainstream or vocational education (eg, pupil referral units and further education colleges). Individual students with special educational needs in mainstream classrooms were excluded at the discretion of teachers as the intervention materials had not been developed for use with this population.

Participants were eligible students in the randomised schools, who consented to participate. Opt in consent was obtained from school head teachers/principals before randomisation. Opt-out consent from participants and their parents/guardians was obtained after randomisation. No schools withdrew from the trial and no pupils or parents/carers withdrew consent. Data were collected under examination-like conditions on school premises.

### Randomisation and blinding
Schools were randomly assigned (1:1) to receive STAMPP or alcohol EAN before baseline data were collected. Randomisation was performed by an independent statistician blinded to the identity of the schools. All schools were stratified on Free School Meal Provision (FSM; low/moderate/high), which was taken as a proxy for socioeconomic status. Schools in NI were also stratified by school type (male/female/coeducational).

Schools, students, intervention trainers and delivery staff (teachers) were not blinded to study condition. Data collection was undertaken by a team of researchers that included the trial manager and research assistants, some of whom were not blinded to study condition.

Data analysis of primary and secondary outcomes was undertaken by the trial statistician who was blinded to the study condition.

### Procedures
STAMPP combined a school-based skills development curriculum and a brief parental intervention designed to support parents in setting family rules around drinking

**Table 1** Stages in the STAMPP trial

| Stage | Description |
|---|---|
| Recruitment of schools | ► Schools in Glasgow Local Authority (n=30) were recruited as a complete group following negotiations with education services.<br>► Schools in Inverclyde (n=5) were recruited following a meeting with the head teachers/principals to discuss the practicalities of the trial.<br>► Schools in Northern Ireland (n=70) were recruited individually in the following process: letter of information; follow-up telephone call; individual meeting with head teacher/principals; agree yes/no. |
| Training of teachers | ► One-day training events were held in each study site before both phases of delivery of the classroom component. Training for the following academic year (from September onwards) took place in the preceding June.<br>► Training involved lectures on alcohol (eg, effects of alcohol use, prevalence rates and risk and protective factors for alcohol use), sharing experiences on previous delivery of the programme and skills rehearsal for each of the SHAHRP lessons.<br>► Training involved examination of each of the SHAHRP lessons that covered: myths about alcohol; units of alcohol; reasons why people do/don't drink; alcohol and the body; consequences of 'levels' of drinking; blood alcohol concentration; social and personal harms; alcohol policy; alcohol and the media; advice for teenagers; a 'night out'; pressures faced by young drinkers; and scenario-based discussion.<br>► Each lesson was scheduled to last one lesson period (approximately 40 min) and delivered once a week.<br>► Teachers were provided with support materials (CD-ROMS and workbooks) at each training session to help implement the lessons. |
| Intervention period | ► The intervention period was September–November in both academic years. Phase one involved six lessons and phase two involved four lessons. Schools were asked to complete all lessons within the 3-month delivery window in both phases.<br>► The Parental Brief Intervention coincided with delivery of phase two when the children were in their third year of secondary school and took place in the evening on intervention school premises. The intervention included a brief presentation on the UK Chief Medical Officers' guidelines on alcohol use by young people and a discussion on setting family rules on alcohol. All intervention student parents, regardless of whether they had attended the evening or not, were mailed a leaflet that reinforced these points a few weeks after the parental session.<br>► Final data collection for the primary outcome took place 1 year after all elements of the intervention had been delivered. |

SHAHRP, School Health and Alcohol Harm Reduction Project; STAMPP, Steps Towards Alcohol Misuse Prevention Programme.

(see table 1 for overview of the intervention). The classroom component of STAMPP was based on the SHAHRP intervention and culturally adapted for the settings of delivery.[30] It combined skills training, education and activities designed to encourage positive behavioural change.[23] See online supplementary materials for more details on the content of each lesson. It was a curriculum-based programme delivered in two phases over a 2-year period. As part of the trial, the first phase was delivered when students were in school year 9/S2 (age 12–13 years), and the second phase was delivered during the subsequent year.

The parental component of STAMPP was developed by the trial team and was based on the programme structure of Koutakis and colleagues[24] and Koning and colleagues.[20 21] The component differed in two main ways to these earlier programmes. First, as part of STAMPP, delivery of a single parental component coincided with the delivery of phase two of the classroom curriculum, whereas in Koutakis and Koning, parents' evenings were held several times over the intervention delivery phase. Second, the session was partly based on guidelines

included in the UK Chief Medical Officers' 2009 guidelines for drinking in childhood.[31] All intervention pupil parents, regardless of whether they had attended the evening or not, were mailed an information leaflet a few weeks after the parental session, which reinforced the discussion points.

The control group participants continued with alcohol EAN within their school. In NI, alcohol-related education is delivered in the context of the Personal Development dimension of Learning for Life and Work,[32] while in Scotland, alcohol education is delivered within the context of Curriculum for Excellence.[33] In both contexts, guidelines are offered to schools; however, the precise nature and duration of EAN is at the discretion of individual school managers. Parents/carers of control students did not receive the STAMPP intervention or materials but may have been exposed to alcohol intervention activities in the community as part of independent provision.

Questionnaires were administered to participants at baseline in June 2012 and at three follow-ups: +12, +24 and +33 months. All students that were present at baseline or joined participating schools prior to delivery of

phase one of the intervention were included in the analyses. Parents/carers were asked to complete a short postal questionnaire, which coincided with delivery of the information leaflet. Alcohol rules were assessed using a 10-item scale to measure the degree to which parents/carers permitted their children to consume alcohol in various situations, such as 'in the absence of parents at home' or 'at a friend's party' ($\alpha$=0.86–0.90).[34] Parental alcohol self-efficacy was assessed using a three-item scale assessing the level of confidence the parent/carer had in their own ability to prevent their child from drinking ($\alpha$=0.67).[35] These data were collected to inform future mediation analysis and are not reported here.

## Outcomes

The study had two primary outcomes at 33 months: (1) the prevalence of self-reported HED drinking in the previous 30 days (HED defined as the consumption of ≥6 units (males)/≥4.5 units (females) on one or more occasions) and (2) the number of self-reported harms (caused by own drinking) in the previous 6 months in students. Prespecified secondary outcomes are described in the online supplementary materials, except for those related to the cost-effectiveness analysis that will be reported elsewhere. The original primary outcome was self-reported frequency of consumption of >5 'drinks' in a single drinking episode. However, concerns arose because it became clear that >5 'drinks' could refer to drinks of different alcohol strength and volume. As the objective of the intervention was to reduce HED, the primary outcome was changed to consumption of ≥6 units for males and ≥4.5 units for females, both are 1.5 times the Chief Medical Officer's maximum daily guideline for adults,[31] and this was ratified by the independent Study Steering Committee. This change was implemented before the final wave of data collection, before unblinding, and before any analysis of trial outcome measures at any data collection point had been undertaken.

To assess the HED primary outcome, participants were presented with pictorial prompts of how much alcohol ≥6/≥4.5 UK units represents. Pictures presented the most popular drinks consumed in the two study areas and respondents were asked to report the frequency of consuming this amount of alcohol over the previous month. Harms associated with own use of alcohol were measured using a 16-item scale developed for the Australian SHAHRP trial (internal consistency 0.9).[36] Participants were asked to indicate on a Likert scale how many times in the past 6 months they had experienced the individual harm. For example, participants were asked to report frequency of having a hangover after drinking or if they had got into a physical fight when drinking.

## Statistical analysis

It was calculated that a sample size of 90 schools (45 per study arm; 80 students per school) would be powerful enough (80%; $\alpha$=0.05; (Intra-class correlation) ICC=0.09 based on data from the Belfast Youth Development Study[37]) to detect a standardised effect size of $\delta$=0.2 or a 10% absolute reduction in risk (51% vs 41%) for the primary outcome of HED. Assuming 20% attrition within each cluster (from 100 to 80 students), the target sample size was 90 schools and 9000 students at baseline.

Summary statistics on school and student recruitment, withdrawal and dropout were collated for both trial arms and reported as a participant flow diagram for reporting of cRCT (figure 1). Outcome measure scores from the questionnaires were summarised and tabulated for the trial arms.

The outcome analysis was an intention-to-treat (ITT) analysis using the Complete Case population such that all cases were assessed regardless of intervention and intervention dosage. Logistic regression models estimated the association between STAMPP and the odds of self-reported HED. Negative binomial regression models estimated the association between STAMPP and the number of ARH. All models included school-level random intercepts to account for correlation due to clustering of students within schools. All models adjusted for factors used to stratify randomisation and the outcome's corresponding value at baseline. For details of analysis of secondary outcomes, please see the online supplementary materials. For each primary and secondary outcome, a statistically significant result was concluded if the P value for the treatment arm explanatory variable was <0.025.

Sensitivity analyses included repetition of the primary outcome analysis using the ITT population with different missing data models. These included a 'best case' (missing set to non-HED), 'worst' case (missing set to HED), 'conservative case' (missing in control arm set to non-HED, missing in intervention arm set to HED) and multiple imputation with 50 imputed data sets.

To explore differential intervention effects on the primary measures, prespecified interaction terms were fitted between trial arm and baseline measures thought to predict the effect of intervention on primary outcomes. These were: age (months) at baseline; gender; socioeconomic status (proportion of students in receipt of FSM tertile split); alcohol use behaviour at baseline—age of initiation, use of alcohol in the year prior to baseline, context of use (abstainer/supervised/unsupervised); and in NI, grammar/secondary school.

Process outcomes were assessed across eight prespecified domains (including intervention acceptability and assessment of the content of EAN), using nine data sources. Methodologies included focus groups with students, an online survey with teachers and interviews with senior school staff and stakeholders. Fidelity and completeness of delivery were assessed using bespoke tools and calculation of participation rates at the parent/carer evening.

Data cleaning, data management and preliminary analysis were undertaken using IBM SPSS V.20+. Mplus 7.11 was used for all analyses and Stata/IC V.12.0 was used to verify Mplus models and generate ORs.

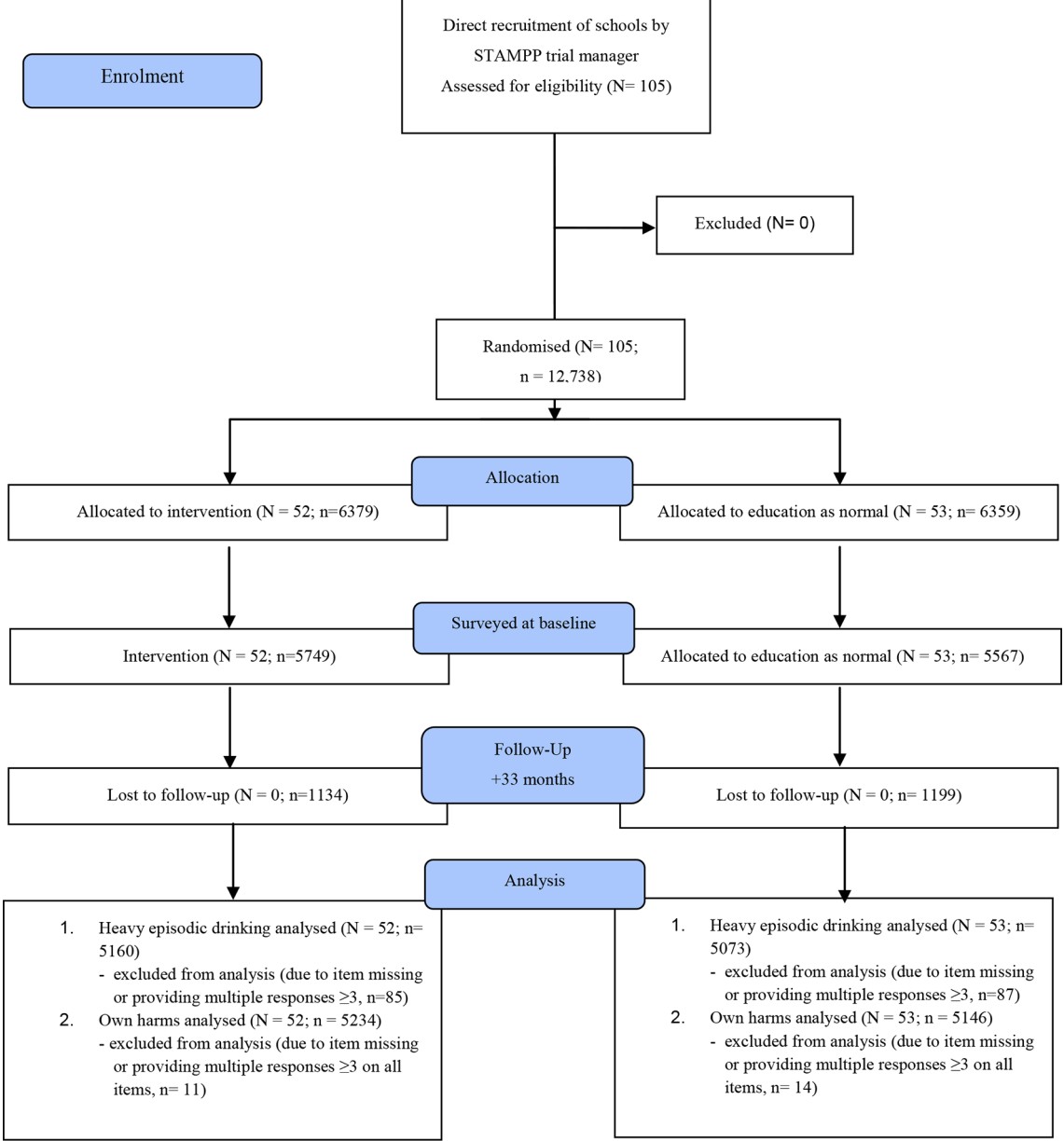

**Figure 1** School and participant flow diagram: STAMPP Trial. Analysis was conducted at 33 months on students who had completed each of the primary outcome measures. N=number of schools; n=student numbers. STAMPP, Steps Towards Alcohol Misuse Prevention Programme.

The trial was registered, number ISRCTN47028486.

### Ethics approval and consent to participate

Participants were eligible students in the randomised schools, who consented to participate. Consent was obtained from school head teachers/principals before randomisation. Consent was obtained from participants and their parents/guardians after randomisation. This was through an opt-out method as opt-in written consent was not required by the ethics committee.

### RESULTS

Figure 1 shows participant flow through the trial. School recruitment began in November 2011 and ended in January 2012. As this was a cRCT of an intervention taking place across several years, student numbers refer to those who completed the questionnaire at each data collection period. No participant or parent/carer requested data were retrospectively removed from analysis. Multiple data collection 'mop up' visits were undertaken with schools, and attrition represents students who were absent on data collection days rather than formal drop out. Of the full sample (those who completed a questionnaire at either baseline or 12 months, n=12 738), 10 405 also completed the questionnaire at 33 months (81.7%). There was a higher attrition rate among students who were male (19.0%), in receipt of FSM (25.8%), and had used alcohol at baseline (25.4%). There was little difference

**Table 2** Baseline characteristics of students according to study condition

| | Control n (%valid) | Intervention n (%valid) |
|---|---|---|
| Total (n=11 316) | 5567 (49.2) | 5749 (50.8) |
| Gender | | |
| Male | 2787 (51.1) | 2834 (50.0) |
| Female | 2670 (48.9) | 2829 (50.0) |
| Missing | 110 | 86 |
| Free school meals | | |
| No | 4289 (77.3) | 4436 (77.5) |
| Yes | 1258 (22.7) | 1290 (22.5) |
| Missing | 20 | 23 |
| Location | | |
| NI | 3469 (62.3) | 3554 (61.8) |
| Scotland | 2098 (37.7) | 2198 (38.2) |
| Missing | 0 | 0 |
| Heavy episodic drinking* | | |
| No | 5082 (92.2) | 5261 (92.4) |
| Yes | 432 (7.8) | 431 (7.6) |
| Missing | 53 | 57 |
| Ethnicity | | |
| White | 4492 (95.3) | 4495 (94.5) |
| Non-white | 248 (4.5) | 293 (5.5) |
| Missing | 827 | 961 |

The percentages are calculated on the basis of the complete cases only.

*Assessed at baseline as consuming >5 drinks in one or more episodes in the last 30 days.

**Table 3** Primary outcomes at 33 months by study group

| | Unadjusted results | | Adjusted model results | |
|---|---|---|---|---|
| | Control N (%valid) | Intervention N (%valid) | OR/IRR | 95% CI |
| HED (frequency) | | | | |
| None | 3773 (74.4) | 4281 (83.0) | 0.60 | 0.49 to 0.73 |
| One or more occasion | 1300 (25.6) | 879 (17.0) | | |
| Missing | 1286 | 1219 | | |
| ARH (frequency) | | | | |
| None | 3126 (60.7) | 3408 (65.1) | 0.92 | 0.78 to 1.05 |
| One or more occasion | 2020 (39.3) | 1826 (34.9) | | |
| Missing | 1213 | 1145 | | |
| Median (IQR) | 0 (2) | 0 (3) | | |

ARH, alcohol-related harm; HED, heavy episodic drinking; IRR, incidence rate ratio.

in attrition between the control and intervention arms of the trial (around one percentage point difference). Attrition also varied by location, with a higher rate in Scotland (24.0%) compared with NI (15.0%). Across schools, attrition varied from 1.5% to 32.0%. There were no unintended harms or adverse effects reported.

Baseline data collection took place in June 2012 with the following follow-up data collection points: 12 months (after delivery of phase one of the classroom component); 24 months (after delivery of the parental intervention and phase two of the classroom component); and 33 months. The trial ended as planned after final data collection and analysis.

Baseline characteristics of students (n=11 316) are presented in table 2. No significant differences in baseline characteristics were detected between control and intervention arms. Overall parental/carer participation was low. A total of 319 parent(s)/carer(s) attended the intervention evenings in NI (9% of those eligible) and 63 parents attended in Scotland (2.5%). With respect to the follow-up mailed intervention, 1074 returns were received from parent(s)/carer(s) in NI (a 31% return) and 440 in Scotland (18%).

Table 3 shows the count and percentages of respondents reporting drinking above the primary outcome threshold (≥6/≥4.5 units) at 33 months and the adjusted model results by study arm (OR; incidence rate ratio (IRR)). Around one in five participants reported at least one episode in the last 30 days. The prevalence of episodes was around 9 percentage points higher in the control group (26%) than in the intervention group (17%). Taking the within (pupil) level variance (fixed at 3.29) and the between (school) level variance (0.454 for the full sample), estimated using a null two level model, the corresponding ICC for the full sample was 0.121. See online supplementary tables S1 and S2 show the full random intercept models for the primary outcomes at 33 months.

Figure 2 displays the count of respondents reporting ARH at 33 months by study group. Around two-thirds of students (63%) reported no ARHs. The median number of harms was equivalent in each study arm (0), while

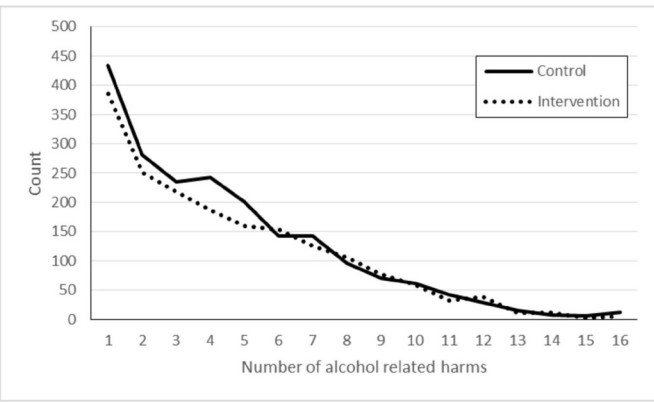

**Figure 2** Count of school children reporting one or more alcohol-related harms by study arm.

the IQR was smaller in the intervention arm than in the control arm (IQR=2 and 3, respectively).

At the school level, the parameter estimates were significant for the intervention arm (estimate=−0.516, SE=0.102; P<0.001). Schools in the intervention arm had lower levels of HED (their intercepts) than those in the control arm (OR=0.596, 95% CI 0.490 to 0.725). This represents a significant intervention effect. However, with respect to ARH, the intervention indicator was non-significant suggesting no difference between the intervention and control schools (estimate −0.101, SE=0.083; P=0.222; IRR=0.916, 95% CI 0.780 to 1.052). Across three of the sensitivity analysis models (best case, worst case and multiple imputed data models), the intervention arm coefficient remained significant and retained the same sign for HED (ie, being a school in the intervention arm was associated with having a lower intercept), while ARH remained non-significant. The only exception was the conservative case model, where both primary outcomes were non-significant.

When the primary measures were assessed at +24 months, as secondary outcomes, the intervention arm was significant at a 0.05 level (β=−0.241; P=0.041) in the HED model but failed to reach the much stricter threshold used within this study (P<0.025) (online supplementary table S3). The intervention arm was also non-significant when the ARH outcome was assessed at +24 months (β=−0.144; P=0.22) (online supplementary table S3). In all the other secondary outcomes, including those assessed at +33 months (online supplementary table S4) and at +24 months (online supplementary table S5), the intervention arm was non-significant.

## DISCUSSION

In a large cRCT, we found that the STAMPP intervention reduced self-reported HED in the past 30 days at 33-month follow-up from baseline, compared with EAN, but not ARH associated with own drinking. There were no clear or consistent effects identified in planned secondary or subgroup analyses (age, gender, SES, alcohol use at baseline and location (Scotland vs NI)). It is possible that longer term follow-up and/or emphasis on those drinking might reveal such effects, especially with regard to self-reported ARH, which were low in both control and intervention students. The intervention was well received by both pupils and teachers.

Key strengths of the trial were the large sample size (schools and students), low rates of attrition (no schools dropped out) and relatively high rates of matched data (>80%) across survey waves. This means that the analyses were sufficiently powered. There also appeared to be no comparator bias, as monitoring of delivery of EAN in intervention schools showed that this did not include alcohol education. A major limitation of the work was the failure to attract parents/carers to the brief intervention evening, despite the support of many of the schools. Although all intervention students received a mailed

follow-up leaflet that reinforced the main messages of the parental intervention, relatively low rates of return of the parental questionnaire suggest that only a minority may have read the mailed information. In contrast, parental participation in the structurally similar (ie, classroom and parental components) Swedish Örebro Prevention Program, and the Dutch (PAS alcohol prevention programmes were relatively high.[24 38] Because we chose a parental intervention based on one with face-to-face contact,[21] we attempted to engage parents at school-based meetings. However, it is possible that the use of a DVD or the creation of a web-based presentation could have served this purpose equally well.[22] Universal interventions such as STAMPP require a range of recruitment strategies as there will be different barriers to, and facilitators of, attendance in parental/carer-based actions. Research is therefore needed to assess the relative efficacy of recruitment strategies such as incentives, mass media campaigns, the removal of barriers to attendance (eg, providing transport and childcare) and the use of key community recruiters (influential individuals and organisations).[39] Furthermore, it is also important to understand if some parent/carer subgroups (eg, differentiated on child drinking risk) are more likely to respond to particular recruitment strategies and if this will lead to recruitment biases.

Although we conducted an ITT analysis that helped to preserve sample size, the achieved participation rates are likely to reflect parental/carer attendance in routine UK practice.[40–42] This meant that we were unable to draw any confident inferences about the combined impact of the school and parental intervention (compared with ref 29) or the relative contribution of each component. In practical terms, this means that although the analysis presumed delivery of the combined intervention, discussions with stakeholders about research findings and future delivery are likely to focus on the classroom component (ie, culturally adapted SHAHRP). However, it is noteworthy that in the PAS programme,[21] the classroom component alone did not produce changes in alcohol use behaviours, and these were only observed in pupils receiving the combined intervention. Subsequent mediation analysis of trial data suggested that reduced rate of frequency of drinking or weekly drinking was mediated by changes in parental rules and attitudes towards alcohol (ie, more strict rules and attitudes were developed). It is therefore important that similar analyses are undertaken to better understand mediators of behaviour change in STAMPP recipients. Other weaknesses of the study included the lack of blinding in intervention delivery and in some data collectors. It is plausible that lack of blinding in delivery may led to either under-reporting or over-reporting of alcohol use due to social desirability biases, but using an EAN comparator meant that it was not possible to conceal intervention allocation from teachers, who received specialised training and curriculum materials, or pupils, who would typically receive little or no alcohol education in their usual school year. Lack of blinding in

some data collectors may have also led to either under-reporting or over-reporting of alcohol use due to social desirability biases, although the use of standardised data collection scripts mitigated against this.

Our primary outcome assessment relied on self-report, which may have led to inaccurate reporting of alcohol use through memory, social desirability and other biases.[43] Although adolescent self-reported alcohol questionnaires are generally reliable,[44] there may be differences in reliability between early and late adolescence,[20] and studies of recanting in substance use surveys suggest that this may be an understudied bias in prevention research.[37] However, all students received the same questionnaire and pictorial prompts, and the recall period for the primary outcome used in this study was the previous 30 days, and so if bias had existed, this would have been minimal and equivalent across trial arms.

Although the classroom component of STAMPP was based on the SHAHRP programme, we did not detect a decrease in ARH. Previous studies of SHAHRP in Australia and NI using quasiexperimental designs found that decreases in self-reported ARH at 32 months were associated with intervention exposure.[23 30] Differences with the findings of this trial may be related to factors such as methodology, pupil age, changes in the wider drinking culture and public health environment or other unmeasured cohort effects. While there is a relationship between HED in adolescence and health harms,[1] we have planned further exploratory analyses that will investigate ARH, patterns of reporting and subgroup effects in more detail.

Although we are mindful of differences in school autonomy, governance and oversight and acknowledge regional variability in alcohol use behaviours (eg,[5]), we believe that the findings of this trial are likely to be applicable to other geographies. Schools enrolled in the trial were drawn from urban and more rural areas and from across the socioeconomic gradient. Furthermore, subgroup analyses showed that there were no differential intervention effects on the basis of school geography (ie, NI vs Scotland).

## CONCLUSIONS
The results of this large cRCT provide support for the effectiveness of a combined classroom and brief parental intervention for reducing HED, but not ARH, in young adolescents. Effects on ARH may manifest later, but further research would be required to clarify this.

**Acknowledgements** As well as acknowledging the role played by participating schools and school children, the authors would like to acknowledge the support of the following people in this project: Séamus Mullin, Gerry Bleakney, Owen O'Neill (PHANI); Malachy Crudden (CCMS); Maura Kearney and Fergal Doherty (Psychological Services, Glasgow); Kate Watson (Psychological Services, Inverclyde); and John Butcher and Sandy Cunningham (Education Services, Glasgow).

**Contributors** HS had full access to all of the data in the study and takes responsibility for the integrity of the data and the accuracy of the data analysis.

MM wrote the first draft of the manuscript and subsequent versions and submitted the final version; HS was project PI, contributed to the first draft and subsequent iterations of the manuscript and prepared the final version of the manuscript; AP conducted the statistical analysis and contributed to manuscript drafts; AA, DF, JC, LM, PD and SH all contributed to drafts and approved the submission.

**Funding** This trial was funded by the National Institute of Health Research (NIHR) Public Health Research (PHR) programme (project number 10/3002/09). The Public Health Agency of NI and Education Boards of Glasgow/Inverclyde provided some intervention costs. Diageo provided funds to print classroom workbooks for use only in the Glasgow local authority area. Remaining intervention costs were internally funded.

**Disclaimer** The views and opinions expressed therein are those of the authors and do not necessarily reflect those of the NIHR-PHR, NIHR, NHS or the Department of Health. The research and intervention funders had no involvement in intervention design; design and conduct of the study; collection, management, analysis and interpretation of the data; and preparation, review or approval of the manuscript.

**Competing interests** The sponsor university (LJMU) received and administered a payment from the alcohol industry for printing of student workbooks in the Glasgow trial site only. AP reported that he has previously received funding from the European Foundation of Alcohol Research (ERAB) in relation to the development of statistical models for longitudinal data (2008–2010). DF reported that his department has previously received funding from the alcohol industry for unrelated prevention programme training work. HS reported that his department has previously received funding from the alcohol industry (indirectly via the industry funded Drinkaware charity) for unrelated primary research.

**Patient consent** Obtained.

**Ethics approval** The research was approved by Liverpool John Moores University Research Ethics Committee (11/HEA/097).

**Provenance and peer review** Not commissioned; externally peer reviewed.

**Data sharing statement** Availability of data and materials: the datasets generated during and/or analysed during the current study are not yet publicly available due to the authors undertaking additional analyses and follow-on studies but are available from the corresponding author on reasonable request.

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
