## [Reviewer comments · BMJ Open]

ARTICLE DETAILS

TITLE (PROVISIONAL)	Steps towards alcohol misuse prevention programme (STAMPP): a school and community based cluster randomised controlled trial
AUTHORS	McKay, Michael; Agus, Ashley; Cole, Jonathan; Doherty, Paul; Foxcroft, David; Harvey, Séamus; Murphy, Lynn; Percy, Andrew; Sumnall, Harry

VERSION 1 – REVIEW

REVIEWER	Gregor Burkhart European Monitoring Centre for Drugs and Drug Addiction Portugal
REVIEW RETURNED	03-Oct-2017

GENERAL COMMENTS	The paper is extremely clear, fluid and easy to read. It resumes the current evidence and positions its finding very well within other similar or preceding studies. No aspect for discussion (either methodological or conceptual) has been left out in the paper, and the findings are relevant for practice and for future research. I have nothing to add
---

REVIEWER	Nicola Newton National Drug and Alcohol Research Centre, University of New South Wales, Sydney, Australia.
REVIEW RETURNED	12-Oct-2017

GENERAL COMMENTS	Thank you for the opportunity to review this article. Overall, this is an excellent study conducted with rigorous methodology and sophisticated analyses. Participant retention was high and the sample size large. Results suggest that STAMPP was effective in reducing the prevalence of HED in young people over a 33 month f/u. Whilst the authors did not find a reduction in ARH, I agree it is plausible these effects may manifest later given the extremely low rates of harms in the sample. A few minor points below to improve the manuscript. 1. Could the authors please rephrase the information about the sample and retention rates in the abstract to be clearer like in the results section: "Of the full sample (those who completed a questionnaire at either baseline or 12 months, N=12,738), 10,405 also completed the questionnaire at 33 months (81.7%)".2. Could the authors please briefly mention the statistical analyses in the abstract.3. In Line 88 – universal is said twice.
--

	4. Some newer references could be included in the literature review as there has been a growing body of literature in this area recently. Two review papers published this year that would be highly relevant are: Yap et al, Addiction, 2017 & Newton et al, Drug and Alcohol Review, 2017. 5. The primary outcomes on pg 5-6 (HED & ARH) don't align with original published protocol link but do seem to align with the updated protocol from March 2015. Can the authors please clarify? 6. Could the authors please provide more information in Procedures on how many lessons and how long the student program was. An outline of the lessons would also be useful. 7. Could the authors please provide more information on the control school education including average number of lessons, approach to prevention etc. 8. Given the difficulty engaging parents in the current trial, Could the authors suggest potential ways to increase engagement for future. E.g. through the use of technology.
--	---

REVIEWER	Penny Cook University of Salford, UK
REVIEW RETURNED	13-Oct-2017

GENERAL COMMENTS	This paper tests a schools-based alcohol intervention, STAMPP, in a large, cluster randomised trial. It is commendable that this study managed to keep all the schools engaged in the study. There were sufficient participants for use in a robust analysis. It would be useful to know if the control schools were offered the training after the end of the trial. I have put 'N/A' for 'results presented clearly' because I can't find figures 1 or 2. Please could the editor send this to me? Very minor corrections: Line 82 'adult alcohol-related problems' would be better as 'alcohol-related problems in adulthood' Line 56 'program' should be 'programme' Line 38 (page 21), Discussion, the first line is 'In a large cRCT we found that the STAMPP intervention reduced self reported HED in the past 30 days at 33 months follow-up from baseline, compared with EAN, but not ARH associated with own drinking'--I would prefer the acronyms to be spelt out again at the beginning of the new section. It is particularly challenging to read this first line of the discussion because there are so many acronyms.
--

VERSION 1 – AUTHOR RESPONSE

Many thanks for the opportunity to provide a revised version of this manuscript. We have detailed our revisions/comments in bold font below the issues raised.

Comments from the Associate Editor:

This is a NIHR funded trial and I agree with Cook that it is impressive how the researchers managed to keep the schools engaged on such a large scale.

Thank you.

I think the authors could describe the intervention better (perhaps even include a box) and enclose the Tidier checklist if they find it appropriate, but as it stands it is rather vague.

In line with the comments of reviewer #2 we have included a section in the supplementary file called "Intervention Content". This addresses the matter raised.

The trial registry (<http://www.isrctn.com/ISRCTN47028486>) shows there have been many changes to the prespecified primary and secondary outcomes throughout the trial but the authors provide a justification in the text. But I didn't understand why they decided to leave the secondary outcomes to the supplementary materials.

The core secondary outcome analysis (i.e. the two primary outcomes assessed at +24 months) is now presented and discussed in the main paper. The full tables are also presented in the supplementary material.

In terms of leaving material in supplementary, we wish that the main paper focusses specifically on the main outcomes, we feel that this is important as many of those already interested in seeing it are school-based individuals, and individuals working in Local Authorities, in other words, a non-academic, applied audience. Secondly, the secondary findings are all null findings, and it seems inappropriate to congest a manuscript with repeated null findings. Additionally, the journal suggests 4,000 words as the optimal size for a manuscript so as to maximise 'readability'. We have already exceeded that threshold, and would suggest that the inclusion of a range of null findings would adversely impact on 'readability'.

There are many tables across many pages with many secondary outcomes reported that left me really confused. But given there's only 3 prespecified secondary outcomes "as of 17/02/2015", why don't they report them in the body of the paper?

While there are only three secondary research objectives, objective two lists six secondary outcomes (lifetime drinking, last year drinking, last month drinking, number of drinks in a typical session, age of onset and unsupervised drinking), each one assessed at +33 months and +24 months. All the secondary outcome models reported null findings for the treatment arm. This is now reported in the main paper. The full tables are also presented in the supplementary material. We have also included some material on sensitivity analyses in the main document.

What are the policy implications and public health implications of these findings? I didn't see a participant flowchart (item 13a and b in the CONSORT extension for cluster RCTs).

The participant flow chart has been included as Figure 1. The implications are dealt with in the discussion section.

Reviewer: 1

Reviewer Name: Gregor Burkhart

Institution and Country: European Monitoring Centre for Drugs and Drug Addiction, Portugal

Please state any competing interests: None declared

Please leave your comments for the authors below

The paper is extremely clear, fluid and easy to read. It resumes the current evidence and positions its finding very well within other similar or preceding studies. No aspect for discussion (either methodological or conceptual) has been left out in the paper, and the findings are relevant for practice and for future research. I have nothing to add

We thank the reviewer for his comments.

Reviewer: 2

Reviewer Name: Nicola Newton

Institution and Country: National Drug and Alcohol Research Centre, University of New South Wales, Sydney, Australia.

Please state any competing interests: None declared.

Please leave your comments for the authors below

Thank you for the opportunity to review this article. Overall, this is an excellent study conducted with rigorous methodology and sophisticated analyses. Participant retention was high and the sample size large. Results suggest that STAMPP was effective in reducing the prevalence of HED in young people over a 33 month f/u. Whilst the authors did not find a reduction in ARH, I agree it is plausible these effects may manifest later given the extremely low rates of harms in the sample. A few minor points below to improve the manuscript.

1. Could the authors please rephrase the information about the sample and retention rates in the abstract to be clearer like in the results section: "Of the full sample (those who completed a questionnaire at either baseline or 12 months, N=12,738), 10,405 also completed the questionnaire at 33 months (81.7%)".

This has been done.

2. Could the authors please briefly mention the statistical analyses in the abstract.

This has been done.

3. In Line 88 – universal is said twice.

This has been amended.

4. Some newer references could be included in the literature review as there has been a growing body of literature in this area recently. Two review papers published this year that would be highly relevant are: Yap et al, *Addiction*, 2017 & Newton et al, *Drug and Alcohol Review*, 2017.

These have been included in the literature review as suggested. We have also included references to the Scottish and Irish Curricula.

5. The primary outcomes on pg 5-6 (HED & ARH) don't align with original published protocol link but do seem to align with the updated protocol from March 2015. Can the authors please clarify?

The final changes to protocol were approved by the independent Study Steering Committee and researcher funder in April 2014. This clarified the final primary outcome measures as:

- To ascertain the effectiveness and cost-effectiveness of STAMPP in reducing alcohol consumption (defined as self-reported consumption of ≥ 6 units in a single episode in the previous 30 days for males and ≥ 4.5 units for females) in school pupils (school year 9 or S2 in the academic year 2012/2013) at + 33 months (T3) from baseline. This will be dichotomised at never/one or more occasions'; and

- To ascertain the effectiveness of STAMPP in reducing alcohol-related harms as measured by the number of self-reported harms (harms caused by own drinking) in second-form pupils (school year 9 or S2 in the academic year 2012/2013) at + 33 months (T3) from baseline'.

We can confirm that this change took place prior to commencement of the statistical analysis.

6. Could the authors please provide more information in Procedures on how many lessons and how long the student program was. An outline of the lessons would also be useful.

We have provided this information in the supplementary document entitled "Intervention Content".

7. Could the authors please provide more information on the control school education including average number of lessons, approach to prevention etc.

This is not an easy point to answer. Essentially schools are required to deliver education in Northern Ireland and Scotland under different sets of guidance. In both sites schools are required to deliver

alcohol-related education, although the nature and duration of that education is at the discretion of individual school Head teachers. We have clarified this (with references) in the methods section.

8. Given the difficulty engaging parents in the current trial, Could the authors suggest potential ways to increase engagement for future. E.g. through the use of technology.

We have addressed this issue in the discussion section.

Reviewer: 3

Reviewer Name: Penny Cook

Institution and Country: University of Salford, UK

Please state any competing interests: One of the authors, Harry Sumnall, is an independent member of the study steering committee for an NIHR grant for which I am principal investigator

Please leave your comments for the authors below

This paper tests a schools-based alcohol intervention, STAMPP, in a large, cluster randomised trial. It is commendable that this study managed to keep all the schools engaged in the study. There were sufficient participants for use in a robust analysis. It would be useful to know if the control schools were offered the training after the end of the trial.

Yes schools were offered training and supplied with a number of sets of workbooks. This was part of the agreement for their being involved.

I have put 'N/A' for 'results presented clearly' because I can't find figures 1 or 2. Please could the editor send this to me?

Very minor corrections:

Line 82 'adult alcohol-related problems' would be better as 'alcohol-related problems in adulthood' This has been amended.

Line 56 'program' should be 'programme' This has been amended.

Line 38 (page 21), Discussion, the first line is 'In a large cRCT we found that the STAMPP intervention reduced self reported HED in the past 30 days at 33 months follow-up from baseline, compared with EAN, but not ARH associated with own drinking'--I would prefer the acronyms to be spelt out again at the beginning of the new section. It is particularly challenging to read this first line of the discussion because there are so many acronyms.

We have spelled out the acronyms in this section as requested.

VERSION 2 – REVIEW

REVIEWER	Penny Cook University of Salford, UK
REVIEW RETURNED	20-Dec-2017

GENERAL COMMENTS	The authors have revised the manuscript in line with the previous reviewers' comments. I have spotted a couple of typos: Line 164 page 7 , change 'data was' to 'data were'. near bottom of page 12, 'This data was... and is not reported...' to
---

	'These data were... and are not reported...'
--	--

REVIEWER	Nicola Newton National Drug and Alcohol Research Centre, University of New South Wales, Australia.
REVIEW RETURNED	07-Jan-2018

GENERAL COMMENTS	I am satisfied with the revisions the authors have made and would now recommend the paper for publication. Regarding control school education, in future trials the investigators could include a log for control schools to complete in regards to the number of lessons they delivered and content of the lessons etc.
--